# Adjunctive Probiotics Alleviates Asthmatic Symptoms via Modulating the Gut Microbiome and Serum Metabolome

Ailing Liu,[d] Teng Ma,[a,b,c] Ning Xu,[d] Hao Jin,[a,b,c] Feiyan Zhao,[a,b,c] Lai-Yu Kwok,[a,b,c] Heping Zhang,[a,b,c] Shukun Zhang,[e] Zhihong Sun[a,b,c]

[a]Inner Mongolia Key Laboratory of Dairy Biotechnology and Engineering, Inner Mongolia Agricultural University, Hohhot, Inner Mongolia, China
[b]Key Laboratory of Dairy Products Processing, Ministry of Agriculture and Rural Affairs, Inner Mongolia Agricultural University, Hohhot, Inner Mongolia, China
[c]Key Laboratory of Dairy Biotechnology and Engineering, Ministry of Education, Inner Mongolia Agricultural University, Hohhot, Inner Mongolia, China
[d]Department of Pulmonary and Critical Care Medicine, Weihai Municipal Hospital, Cheeloo College of Medicine, Shandong University, Weihai, Shandong, China
[e]Department of Pathology, Weihai Municipal Hospital, Cheeloo College of Medicine, Shandong University, Weihai, Shandong, China

Ailing Liu, Teng Ma, Ning Xu, and Hao Jin contributed equally to this article. Author order was determined by drawing straws.

**ABSTRACT** Asthma is a multifactorial disorder, and microbial dysbiosis enhances lung inflammation and asthma-related symptoms. Probiotics have shown anti-inflammatory effects and could regulate the gut-lung axis. Thus, a 3-month randomized, double-blind, and placebo-controlled human trial was performed to investigate the adjunctive efficacy of probiotics in managing asthma. Fifty-five asthmatic patients were randomly assigned to a probiotic group ($n = 29$; received *Bifidobacterium lactis* Probio-M8 powder and Symbicort Turbuhaler) and a placebo group ($n = 26$; received placebo and Symbicort Turbuhaler), and all 55 subjects provided details of their clinical history and demographic data. However, only 31 patients donated a complete set of fecal and blood samples at all three time points for further analysis. Compared with those of the placebo group, co-administering Probio-M8 with Symbicort Turbuhaler significantly decreased the fractional exhaled nitric oxide level at day 30 ($P = 0.049$) and improved the asthma control test score at the end of the intervention ($P = 0.023$). More importantly, the level of alveolar nitric oxide concentration decreased significantly among the probiotic receivers at day 30 ($P = 0.038$), and the symptom relief effect was even more obvious at day 90 ($P = 0.001$). Probiotic co-administration increased the resilience of the gut microbiome, which was reflected by only minor fluctuations in the gut microbiome diversity ($P > 0.05$, probiotic receivers; $P < 0.05$, placebo receivers). Additionally, the probiotic receivers showed significantly changes in some species-level genome bins (SGBs), namely, increases in potentially beneficial species *Bifidobacterium animalis*, *Bifidobacterium longum*, and *Prevotella* sp. CAG and decreases in *Parabacteroides distasonis* and *Clostridiales bacterium* ($P < 0.05$). Compared with that of the placebo group, the gut metabolic potential of probiotic receivers exhibited increased levels of predicted microbial bioactive metabolites (linoleoyl ethanolamide, adrenergic acid, erythronic acid) and serum metabolites (5-dodecenoic acid, tryptophan, sphingomyelin) during/after intervention. Collectively, our results suggested that co-administering Probio-M8 synergized with conventional therapy to alleviate diseases associated with the gut-lung axis, like asthma, possibly via activating multiple anti-inflammatory pathways.

**IMPORTANCE** The human gut microbiota has a potential effect on the pathogenesis of asthma and is closely related to the disease phenotype. Our trial has demonstrated that co-administering Probio-M8 synergized with conventional therapy to alleviate asthma symptoms. The findings of the present study provide new insights into the pathogenesis and treatment of asthma, mechanisms of novel therapeutic strategies, and application of probiotics-based therapy.

Address correspondence to Zhihong Sun, sunzhihong78@163.com, or Shukun Zhang, zhangshukun0475@126.com.

**KEYWORDS** *Bifidobacterium lactis* Probio-M8, randomized placebo-controlled trials, probiotic adjuvant therapy, species-level genome bins, metabolomics, gut-lung axis, asthma

Asthma is a common chronic inflammatory respiratory disease with high morbidity and mortality all over the world, afflicting over 330 million children and adults, and the afflicted population is anticipated to increase to 400 million by 2025 (1). The etiology and pathogenesis of asthma are still not well understood, but the disease is known to relate to multiple factors, including genetics, infection, immunity, nutrition, and environment. Recently, the potential impact of human microbiota in the pathogenesis of asthma and its relation with the disease phenotypes have received special attention (2, 3). The human microbiota community consists of a wide diversity of microbes, including bacteria, archaea, protozoa, fungi, and viruses, among which bacteria and viruses are most widely explored (4). Immune function development and susceptibility to diseases, including asthma, are closely associated with host microbiota diversity and composition (5). Recent experimental evidence has also shown that localized alterations in commensal microbiota or dysbiosis directly affect not only the colonizing body compartment but also distant organs and systems, including the pulmonary system that regulates respiratory health and diseases like asthma, cystic fibrosis, lung cancer, and respiratory infections (6). The latest research in the respiratory immune system has broadened our understanding of bidirectional communications between the gastrointestinal tract and the respiratory system and the crucial role of the resident microbiota in the gut-lung axis (4). For example, mice bred under germfree or sterile conditions showed stronger allergic responses to allergen stimulation due to the lack of gut microbiota (7). Interestingly, the development of respiratory disease-related immune responses was reversible and steerable through target modulation of specific microbiota or application of probiotics (8, 9). There are obvious connections between asthma exacerbation and host microbial communities, especially the gut microbiota, but the mechanisms of how they affect asthmatic symptoms and the serum metabolome, particularly in response to drug treatment, are still largely unknown.

Probiotics are defined as "live microorganisms, which when administered in adequate amounts, confer a health benefit on the host" (10). Members of the *Lactobacillus* and *Bifidobacterium* genera are the most commonly used probiotic bacteria that help prevent and improve allergies and respiratory diseases (11), as many of them can colonize stably in the intestinal tract, modulate gut microbial composition, enhance microbial metabolite levels, mainly short-chain fatty acids (SCFAs), and modulate host immunity (5, 12). Preclinical experiments have shown that targeted gut microbiome manipulation through probiotics has preventive and therapeutic effects on asthma (13). Experimental evidence obtained from rodent models consistently showed beneficial effects of probiotic application to the hosts (14). For example, various *Lactobacillus rhamnosus* strains have shown anti-inflammatory effects in the airway in murine models, suppressing the total immunoglobulin E (IgE) production and pulmonary eosinophil-associated inflammation (15). Other strains, e.g., *Bifidobacterium breve* M-16 (16), *Lactococcus lactis* NZ9000 (17), and *Bifidobacterium lactis* Bb-12 (18), could suppress asthmatic symptoms. On the contrary, unlike in animal experiments, application of probiotics in managing asthma has not achieved consistent results in human clinical trials. Two randomized placebo-controlled trials (RCTs) of pediatric asthma found that ingesting probiotics effectively reduced exhaled nitric oxide concentration, decreased clinical asthma scores, and improved pulmonary function in patients (19, 20). A compound probiotic product that comprised two bacterial strains showed beneficial immunomodulatory activities in adult asthmatic subjects (21). However, a meta-analysis found that probiotics had no obvious therapeutic effect on asthma, while another meta-analysis concluded that taking probiotics in early life helped reduce the IgE level and protect against atopic sensitization but did not seem to protect against asthma and/or wheezing (22, 23). To clarify the controversy clinical effects of probiotics on asthmatic symptoms, future trials should be meticulously designed, including careful selection of strains, application of high-quality RCT procedures

and detailed documentation guidelines, incorporation of longer follow-up periods, and employment of advanced and reproducible microbiota analysis methods (2, 24).

This work described a 3-month RCT that investigated the adjuvant therapeutic effect of *Bifidobacterium lactis* M8 (Probio-M8) in managing asthma in two bronchial asthma cohorts (probiotic group, *n* = 29; placebo group, *n* = 26). Probio-M8 is a probiotic strain isolated from human breast milk of a healthy woman, and the strain has shown various health-promoting properties in our previous studies (25, 26). The objectives of this work were to evaluate the clinical efficacy of Probio-M8 in alleviating asthma and to reveal the mechanisms behind the symptom-attenuating effect by analyzing changes in patients' clinical indices, gut microbiome, and serum metabolome. The findings of the present study would provide new insights into the pathogenesis and treatment of asthma, mechanisms of novel disease management strategies, and application of probiotics-based therapy.

## RESULTS

**Probiotics improved asthma-related symptoms.** To evaluate the adjunctive efficacy of probiotics in managing asthmatic symptoms, several clinical parameters were monitored at days 0, 30, and 90, including the asthma control test (ACT) score, CaNO, peak expiratory flow (PEF), PEV1, and FeNO, and no significant difference was observed in these indicators between the probiotic and placebo groups at day 0. Interestingly, compared to that of the placebo group, co-administering Probio-M8 with Symbicort Turbuhaler significantly decreased the FeNO level at day 30 ($P$ = 0.049) and improved the ACT score at the end of the intervention ($P$ = 0.023). More importantly, the Probio-M8 receivers had significantly reduced CaNO level at day 30 ($P$ = 0.038), and such effect was even more obvious after 90 days of intervention ($P$ = 0.001). In addition, compared with that at day 0, the Probio-M8 receivers exhibited significantly higher ACT scores with reduced CaNO and FeNO levels during/after the intervention ($P$ < 0.05). No adjunctive efficacy was observed in the lung function indicators (PEF, PEV1, PVC) and the peripheral eosinophil count between the Probio-M8 and placebo groups (Table S3). There was no significant difference in the serum IgE level between Probio-M8 and placebo groups at all time points; however, the IgE level of placebo group significantly increased during and at the end of the study ($P$ < 0.003; Fig. 1a). These results suggested that the application of Probio-M8 as adjunctive treatment significantly improved some of the asthma-associated clinical symptoms.

**Features of metagenomes.** In-depth metagenomic analysis was performed on a total of 93 samples from 31 participants collected at days 0, 30, and 90. A total of 508, 430, and 2,601 metagenome-assembled genomes (MAGs) were assigned to partial-, medium-, and high-quality MAGs after the bin refinement (Table S4). A total of 389 SGBs were extracted from the high-quality MAGs (Table S5), and their average mappability was 77.21% ± 3.73% of raw reads/sample. The high mappability indicated that most genomes belonged to already known microbial communities (Fig. 1b). The 389 SGBs were distributed across 10 phyla, 19 classes, 24 orders, 39 families, 78 genera, and 247 species. Sixty-two SGBs were unmappable to the species level, representing the uncultivated species (Fig. S2a), and they belonged mainly to the phylum *Firmicutes* (90.32%). The current data set was then cross-compared with the IGG and Cell_MAGs data sets, and 27 SGBs did not match with any currently known species with <95% average nucleotide similarity, confirming their novelty (Table S5; Fig. S2b).

The deep shotgun sequencing enabled us to track Probio-M8 in the samples by mapping the clean reads of the metagenomic data set to the Probio-M8 reference genome. A gradual increase in the abundance representing Probio-M8 strain was observed in the probiotic but not the placebo group, suggesting that the ingested Probio-M8 strain could easily pass through the digestive tract (Fig. S2c).

**Probiotics regulated gut microbiota composition.** No significant difference was observed in the alpha diversity (represented by the Shannon diversity index) and beta diversity (shown by principal-coordinate analysis [PCoA] and Adonis analysis) of both Probio-M8 and placebo groups at all time points ($P$ > 0.05; Fig. 2a and b). However,

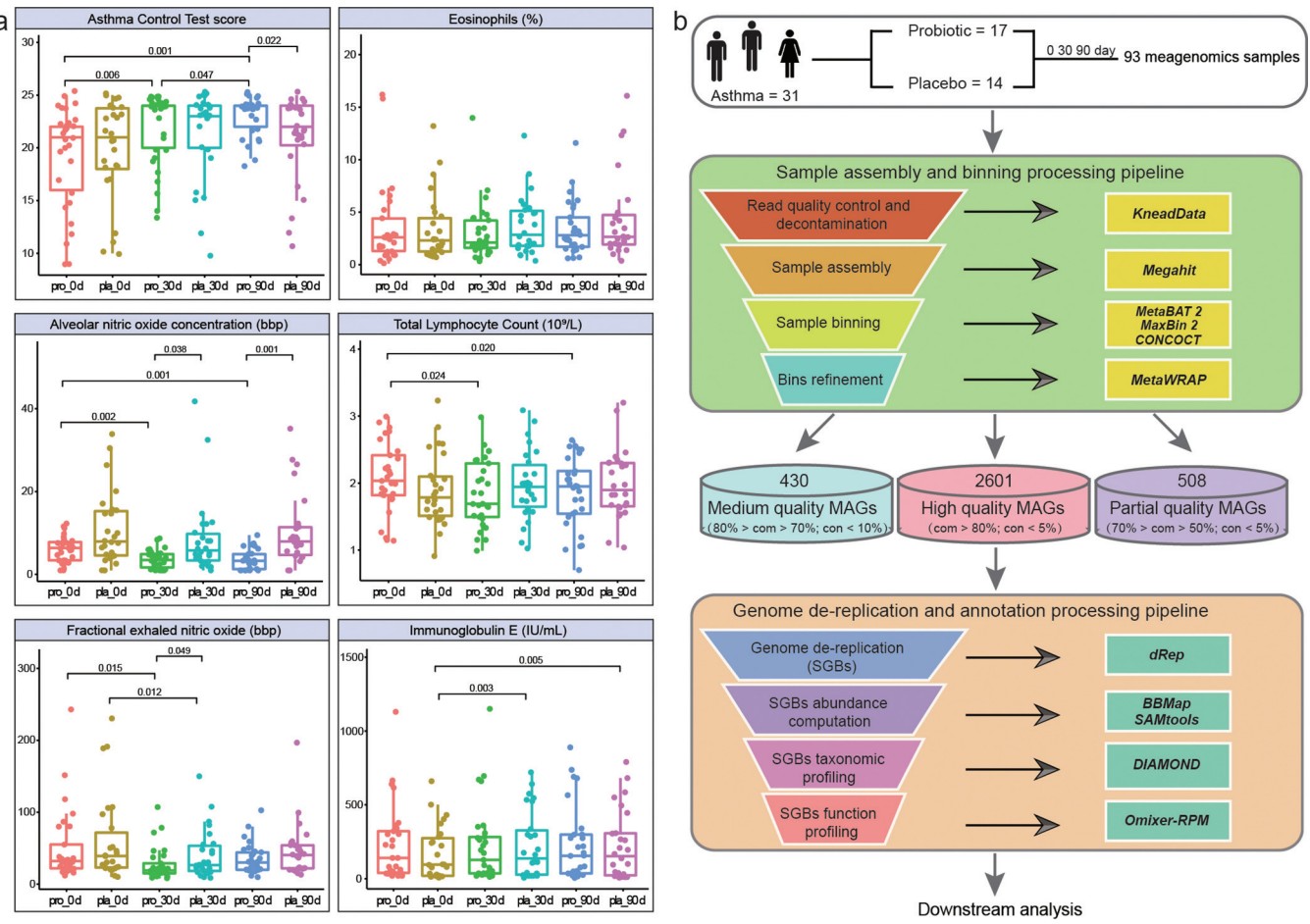

**FIG 1** Clinical indicators of asthma-associated symptoms and metagenomic reconstruction pipeline. (a) Statistical differences in the clinical parameters between groups were evaluated with Wilcox test or *t* test. (b) The workflow of sampling, assembling of metagenome-assembled genomes (MAGs), binning, refinement, and dereplication. A total of 389 high-quality species-level genome bins (SGBs) were identified in the complete data set. com and con represent levels of completeness and contamination, respectively.

the alpha diversity of the placebo group was significantly reduced at days 30 and 90 (*P* values of <0.05 and <0.005) compared with that at day 0, while it remained stable for the Probio-M8 group throughout the trial period (Fig. 2a), suggesting that probiotic intake might help maintain the stability of gut microbiota diversity. In addition, no significant difference was found in the alpha/beta diversity of the gut microbiota at all time points when subgroup comparison by age was performed (adult: 28 to 59; elderly: 61 to 71; *P* > 0.05; Fig. S2d and e).

To further explore changes in the gut microbiota at a finer level during the course of the intervention, responsive SGBs were identified, which were defined as SGBs/species that showed no significant difference in abundance between the two groups at day 0 but became differentially abundant between time points or between treatment groups (Table S6). For the probiotic group, the abundance of three SGBs, T16C.M023 (*Bifidobacterium animalis*), C14B.M022 (*Roseburia hominis*), and C1A.M016 (*Ruminococcus callidus*), increased significantly during/after the intervention period, while one SGB (C5A.M066, representing *Parabacteroides distasonis*) decreased significantly (*P* < 0.05). For the placebo group, 16 responsive SGBs were detected, including C8B.M011, T17A.M023, C12C.M016, T5B.M042, and T16A.M006 (family *Clostridiaceae*), T10B.M022, C6C.M025, and T4C.M019 (phylum *Firmicutes*), and T4C.M040 and C14C.M035 (uncultured bacteria), exhibiting significant changes in abundance during/after the intervention. Three SGBs, C13B.M008, T8B.M046 (genus *Blautia*), and T17B.M018 (*Ruminococcus* sp. AF37-6AT), decreased significantly during/after the intervention, while three SGBs, C1C.M051 (*Coprococcus eutactus*), C14A.M016

Microbiology Spectrum

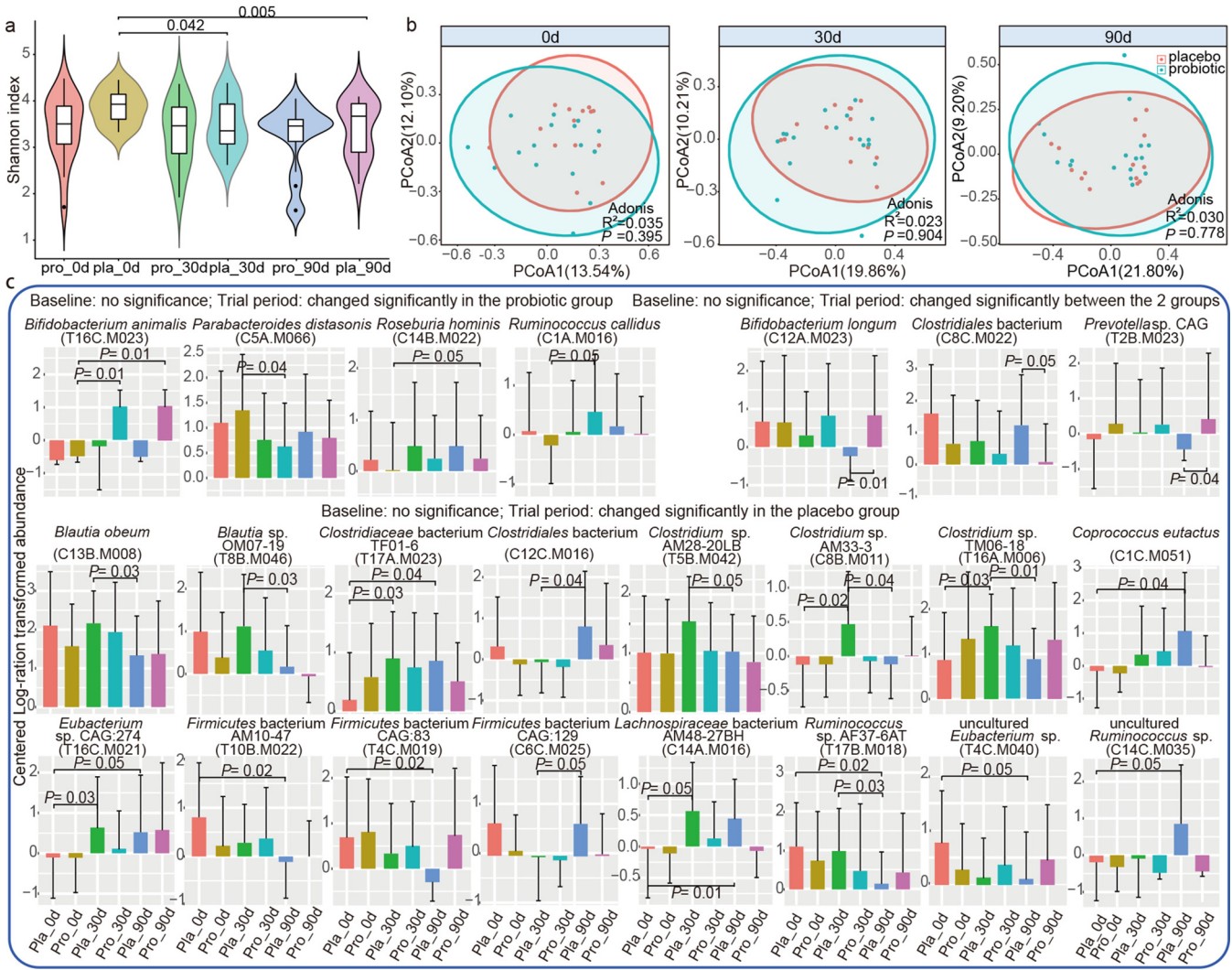

**FIG 2** Microbial diversity and species-level genome bins (SGBs) features of fecal metagenome data set of participants. (a) Shannon diversity index of the probiotic and placebo groups at days 0 (0d), 30 (30d), and 90 (90d). (b) Principal coordinates analysis (PCoA) score plots of the probiotic and placebo groups. Symbols representing samples of the placebo and probiotic groups at different time points are shown in different colors. (c) Significant differential SGBs between probiotic and placebo groups at different time points and the Benjamini-Hochberg procedure-corrected *P* values between sample pairs are shown. A corrected *P* value of <0.05 was considered statistically significant.

(*Lachnospiraceae bacterium* AM48-27BH), and T16C.M021 (*Eubacterium* sp. CAG:274), increased significantly ($P < 0.05$). Three probiotic-responsive SGBs showed differential abundance between groups at the end of the intervention. At day 90, significantly more C12A.M023 (*Bifidobacterium longum*) and T2B.M023 (*Prevotella* sp. CAG) were found in the Probio-M8 group than in the placebo group, while an opposite trend was observed for C8C.M022 (*Clostridiales bacterium*; $P < 0.05$; Fig. 2c).

**Probiotics modulated GMMs and predicted gut bioactive compounds.** Then, probiotic-specific modulation of gut metabolic molecules (GMMs) and predicted gut bioactive compounds were explored. A genome-centric metabolic reconstruction was conducted to discern changes in the GMMs encoded in the 389 SGBs using the MetaCyc and KEGG databases, focusing mainly on the modules related with the development, pathophysiology, and immune responses associated with asthma (Table S7). The identified GMMs belonged to 10 different phyla, mainly *Firmicutes* (72.75%), *Bacteroidetes* (13.11%), and *Actinobacteria* (4.11%). The GMMs involved in acetate synthesis I, quinolinic acid degradation, and polyunsaturated fatty acids (PUFAs) synthesis were distributed to a large number of SGBs (Fig. S2d). Interestingly, the distribution of the most prevalent GMMs was largely different between the placebo and probiotic

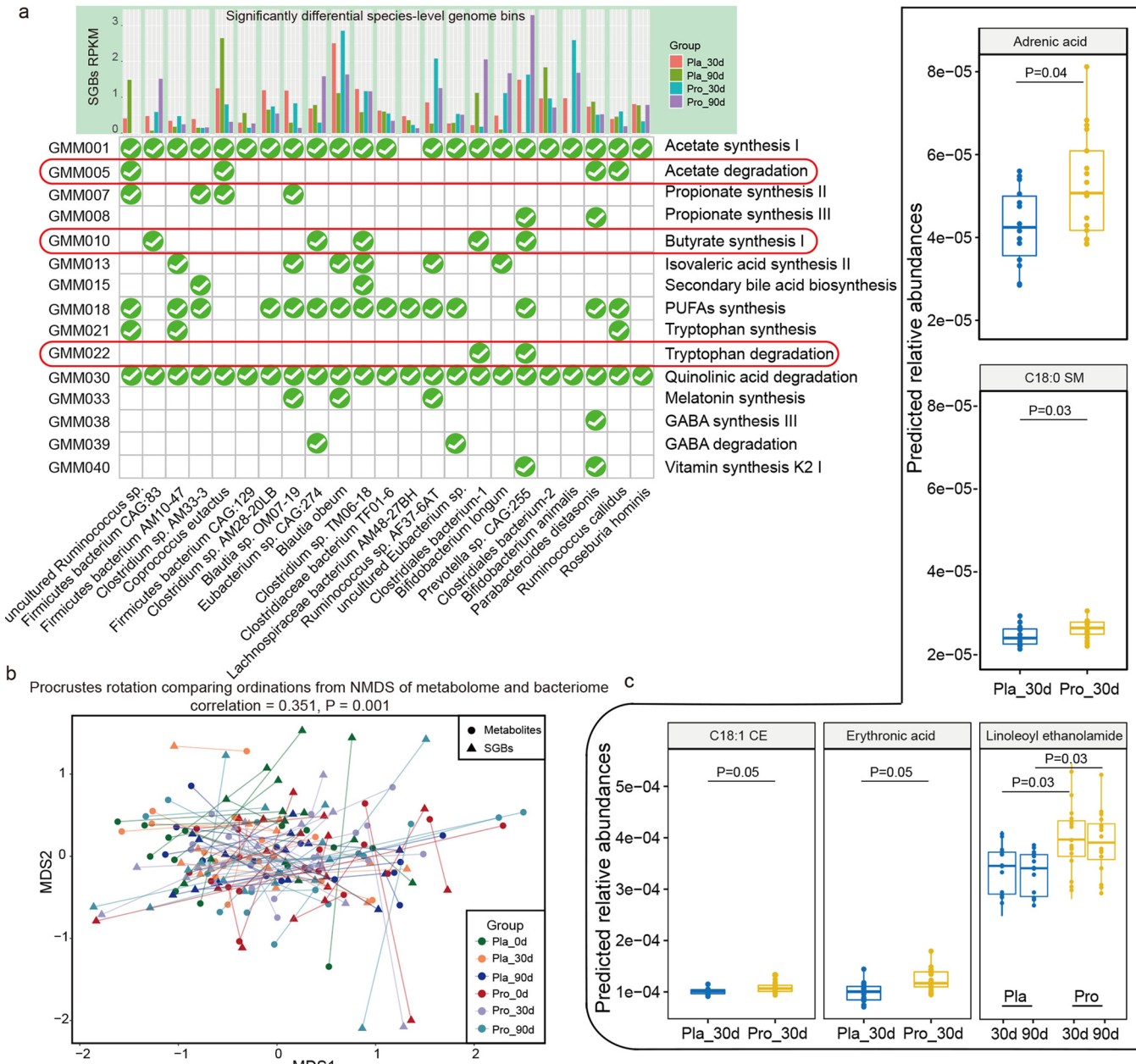

**FIG 3** Changes in the profiles of gut metabolic modules (GMMs) and predicted metabolites at days 30 and 90. (a) Distribution of selected GMMs (relating with asthma development, pathophysiology, and immune system) in significant differential species-level genome bins (SGBs) between the probiotic (Pro) and the placebo (Pla) groups. The check mark circle represents the presence of the specific GMM in the corresponding SGB. (b) Procrustes analyses performed on the predicted microbiomes and metabolomes of the probiotic (Pro) and placebo (Pla) groups at day 0 (0d), day 30 (30d), and day 90 (90d), showing a positive cooperativity between the microbiome and metabolome profiles (correlation = 0.351; $P$ = 0.001). (c) Boxplots showing the contents of predicted differential bioactive metabolites that were responsive to the probiotic adjuvant treatment. Benjamini-Hochberg procedure-corrected $P$ values between sample pairs are shown, and a corrected $P$ value of <0.05 was considered statistically significant.

groups during/after intervention. Two GMMs, butyrate synthesis I and tryptophan degradation, were dominated in the probiotic group, while acetate degradation was highly represented in the placebo group (Fig. 3a).

The gut bioactive compounds were further predicted by MelonnPan, and a total of 80 metabolites were identified. Procrustes analysis is a statistical method that displays multi-omics data sets in low-dimensional space after data dimensionality reduction, and it has been increasingly used in microbiome and metabolomics research to evaluate the similarities and differences between data sets (27). Notably, the results of Procrustes analysis in this study confirmed a good cooperativity between the gut

microbiome and metabolome (correlation = 0.351; $P$ = 0.001), suggesting cohesive changes between the predicted gut metabolites and SGBs during the course of treatment (Fig. 3b). Thirteen responsive metabolites exhibited no significant changes at day 0 but became significant differential features between the placebo and probiotic groups during/after treatment (Table S8). Significantly more adrenic acid, $C_{18}$:0 sphingomyelin, $C_{18}$:1 cholesterol ester, erythronic acid, and linoleoyl ethanolamide were detected in the probiotic group at day 30 or at both days 30 and 90 ($P < 0.05$; Fig. 3c), representing the adjunctive efficacy of Probio-M8 in promoting the synthesis of specific bioactive metabolites and related pathways.

**Probiotics modulated serum metabolites.** To further explore the interplay between the human gut microbiome and host metabolism, serum metabolomic changes were monitored during the course of intervention. Symbols representing the quality control (QC) samples clustered tightly on the principle-component analysis (PCA) score plot, indicating that the liquid chromatography-mass spectrometry (LC-MS) system and conditions were of high stability and that the generated data were valid for downstream analysis (Fig. S2e). The serum metabolomes of the probiotic and placebo groups were marginally different at 30 days ($P < 0.06$). However, when the serum metabolome was longitudinally analyzed by PCoA and Adonis test, it was found that co-administering Probio-M8 with Symbicort Turbuhaler resulted in marginal to significant shifts in the serum metabolomes on day 30 and day 90 ($P$ values of 0.06 and 0.04, respectively; Adonis test), but similar changes were not observed in the placebo group (Fig. 4a). Thirty differential abundant metabolites between the two groups were identified by applying a partial least-squares-discriminant analysis (PLS-DA) model (cutoff variable importance in projection [VIP] > 2; $P < 0.05$; Fig. 4b). The selected features were searched across the Blood Exposome Database, annotating 16 features to the level of metabolites (Table S9). Six of these features, namely, 5-dodecenoic acid, enterodiol, syringic acid, 1-palmitoyl-rac-glycerol, tryptophan, and sphingomyelin, were enriched in the probiotic group compared with those in the placebo group at day 90 (Fig. 4c). These results suggested that co-administering Probio-M8 and Symbicort Turbuhaler regulated some serum metabolites specifically during the course of treatment.

**Probiotics regulated gut virome profiles.** Advanced bioinformatic tools, including VIBRANT and DeepVirFinder, were used to identify bacteriophage sequences in the metagenomes, and 13,078 nonredundant viral OTUs (vOTUs) were detected. The average genome size of these vOTUs was 28.50 ± 56.17 kbp (Table S10), which was remarkably smaller than the bacteriophage genomes found in the RefSeq database (average of 38.5 kbp from around 6,500 viruses), suggesting that most currently detected vOTUs were fragmented or partial genomes. Only 12.73% of them could be assigned to a specific family, suggesting a high novelty of the gut virome (Fig. 5a). No significant difference was found in the alpha and beta diversity of the gut virome between Probio-M8 and placebo groups at all time points ($P > 0.05$; Fig. 5b and c). However, interestingly, the diversity of the gut virome decreased significantly in the placebo ($P < 0.05$) but not in the probiotic group (Fig. 5b), indicating that co-administering probiotics with Symbicort Turbuhaler helped maintain the gut bacteriophage diversity. The PCoA found no significant difference in the overall gut bacteriophage profile between different time points in both placebo and probiotic groups ($P > 0.05$, ANOSIM; Fig. 5c).

Ten bacteriophage families were identified in our dataset, and the most dominant viral families were *Iridoviridae* and *Podoviridae* (Fig. 5d). For the Probio-M8 group, the average abundance of *Myoviridae* decreased significantly at day 30, while that of *Herelleviridae* increased significantly at day 90 (Fig. S3a). Notably, the alpha diversity of the gut bacterial microbiota correlated strongly with that of the gut virome (overall: $R$ = 0.895, $P < 0.001$, Fig. 5e; $R$ = 0.916, $P < 0.001$ for the probiotic group; $R$ = 0.873, $P < 0.001$ for the placebo group). Consistently, Procrustes analysis found a strong cooperative relationship between the gut bacterial microbiota and the gut virome (correlation = 0.782; $P$ = 0.001; Fig. 5f), suggesting a high infectious specificity of gut viruses toward

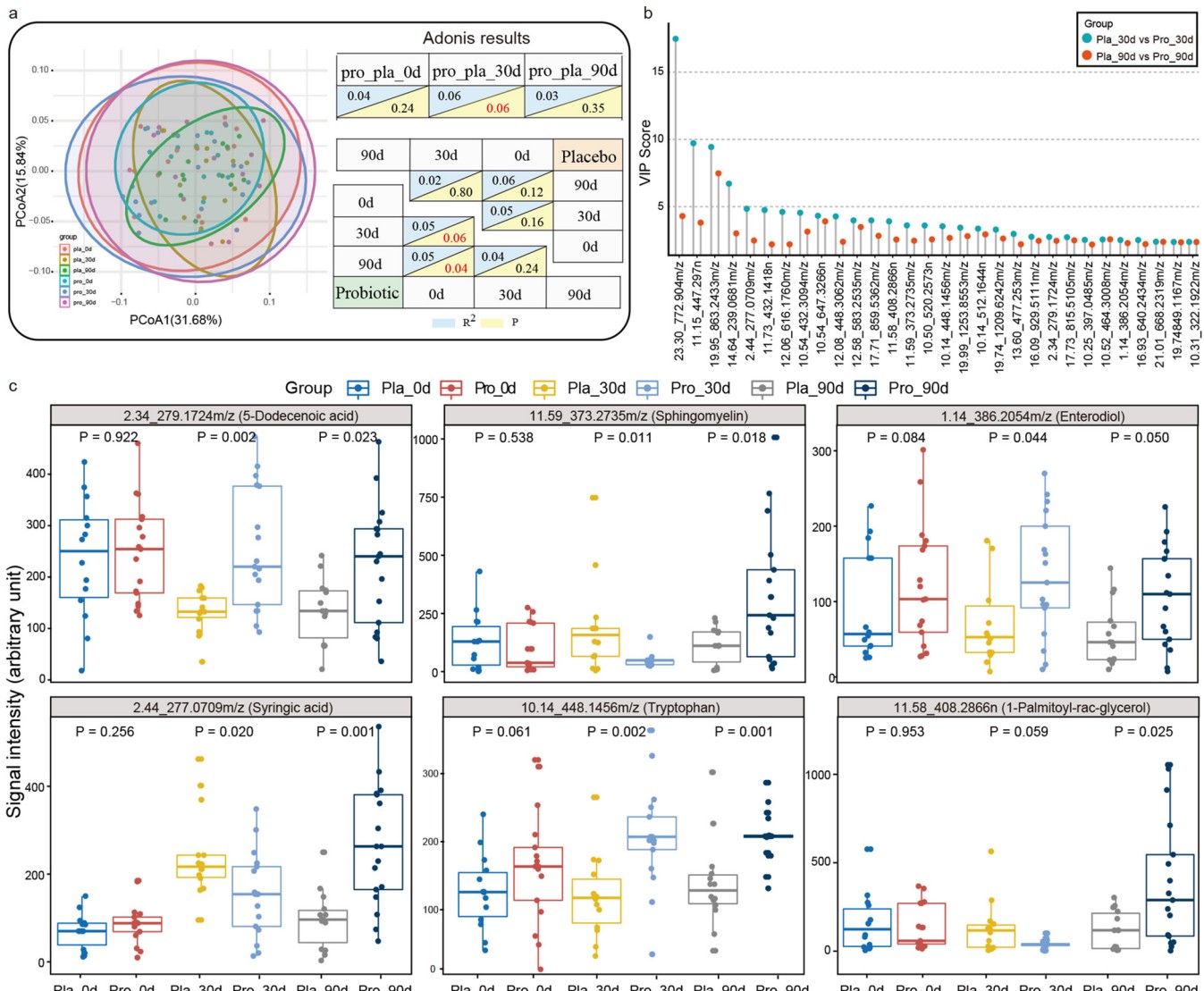

**FIG 4** Changes in serum metabolomes of asthmatic patients during and after the intervention. (a) Principal coordinates analysis (PCoA) score plots and Adonis results of the probiotic (Pro) and placebo (Pla) groups at days 0 (0d), 30 (30d), and 90 (90d). Symbols representing samples of the placebo and probiotic groups at different time points are shown in different colors. Significant differences between groups were evaluated by Adonis test, and P values generated by pairwise comparison are shown. (b) Significant differential metabolites detected by liquid chromatography/mass spectrophotometry (LC/MS) at days 30 (30d) and 90 (90d) between the probiotic (Pro) and placebo (Pla) groups. No significant difference was found in the quantities of these metabolites between the two groups at day 0. The variable importance in projection (VIP) scores were generated by partial least-squares-discriminant analysis (PLS-DA) of metabolomic data sets of the two groups. (c) Box plots showing comparison between levels of serum metabolites that were responsive to the probiotic adjuvant treatment. Benjamini-Hochberg procedure-corrected P values between sample pairs are shown, and a corrected P value of <0.05 was considered statistically significant.

their bacterial hosts. The interactive connection networking between the gut microbiota and gut virome was more robust in the probiotic group than in the placebo group (Fig. S3b and c).

**Effect size of gut microbiota on the monitored features.** To understand how intervention-induced changes in the gut microbiota composition affected each set of the monitored features (including asthmatic clinical indicators, predicted gut bioactive metabolome, serum metabolome, and gut virome) during the course of probiotic adjuvant therapy, a permutational multivariate analysis of variance (PERMANOVA)-based effect size analysis was performed. Our results showed that 272 SGBs ($P < 0.05$) contributed to an average of 55.34% (range from 28.8% to 78.31%) of variances of the included features between the probiotic and placebo groups. Notably, the gut microbiota of the probiotic group explained a larger variance than the placebo group in the

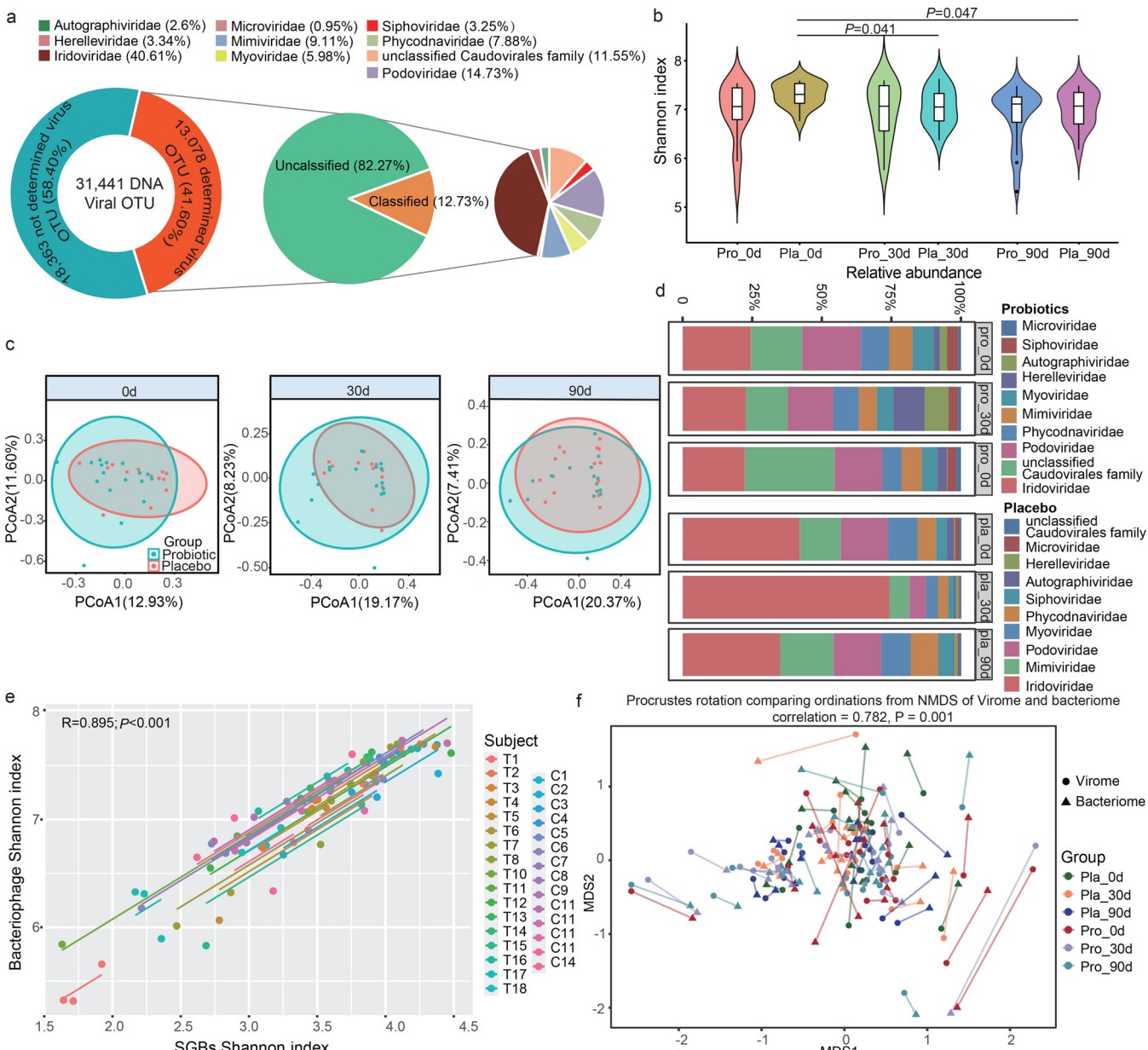

**FIG 5** Changes in the gut virome of asthmatic patients during and after the intervention. (a) Family-level taxonomic distribution of virome metagenome. Our data set contained 31,441 qualified DNA viral operational taxonomic units (vOTUs). (b) Changes in Shannon diversity index of the probiotic (Pro) and placebo (Pla) groups during the course of intervention. *P* values (*t* test) between sample pairs are shown. (c) Principal coordinates analysis (PCoA) of gut virome of the probiotic and placebo groups. Symbols representing samples of the placebo and probiotic groups at different time points are shown in different colors. (d) Family-level distribution of phage features of two sample groups at different time points. (e) Correlation between the value of Shannon diversity index of the gut bacterial microbiota and virome; a strong positive correlation was found (*R* = 0.895; *P* < 0.001). Samples representing the probiotic and placebo recipients are represented by the sample codes "T" and "C," respectively. (f) Procrustes analysis performed on gut species-level genome bins (SGBs) and virome of the probiotic (Pro) and placebo (Pla) groups at day 0 (0d), day 30 (30d), and day 90 (90d), showing a positive cooperativity between the gut bacterial microbiota and virome (correlation = 0.782; *P* = 0.001).

predicted bioactive compounds, clinical index, and serum metabolites by 9.80%, 3.60%, and 1.30%, respectively (Fig. 6a). On the other hand, an opposite trend was observed in the bacteriophage metagenome.

Twenty SGBs consistently showed the strongest effect size in response to Probio-M8 treatment, particularly SGBs representing *Bifidobacterium longum* (C12A.M023), *Firmicutes bacterium* AM10-47 (T10B.M022), *Ruminococcus* sp. AF37-6AT (T17B.M018), and *Firmicutes bacterium* CAG:83 (T4C.M019), which were enriched in the probiotic group at the end of the intervention (Fig. S3d). Taken together, these results suggested

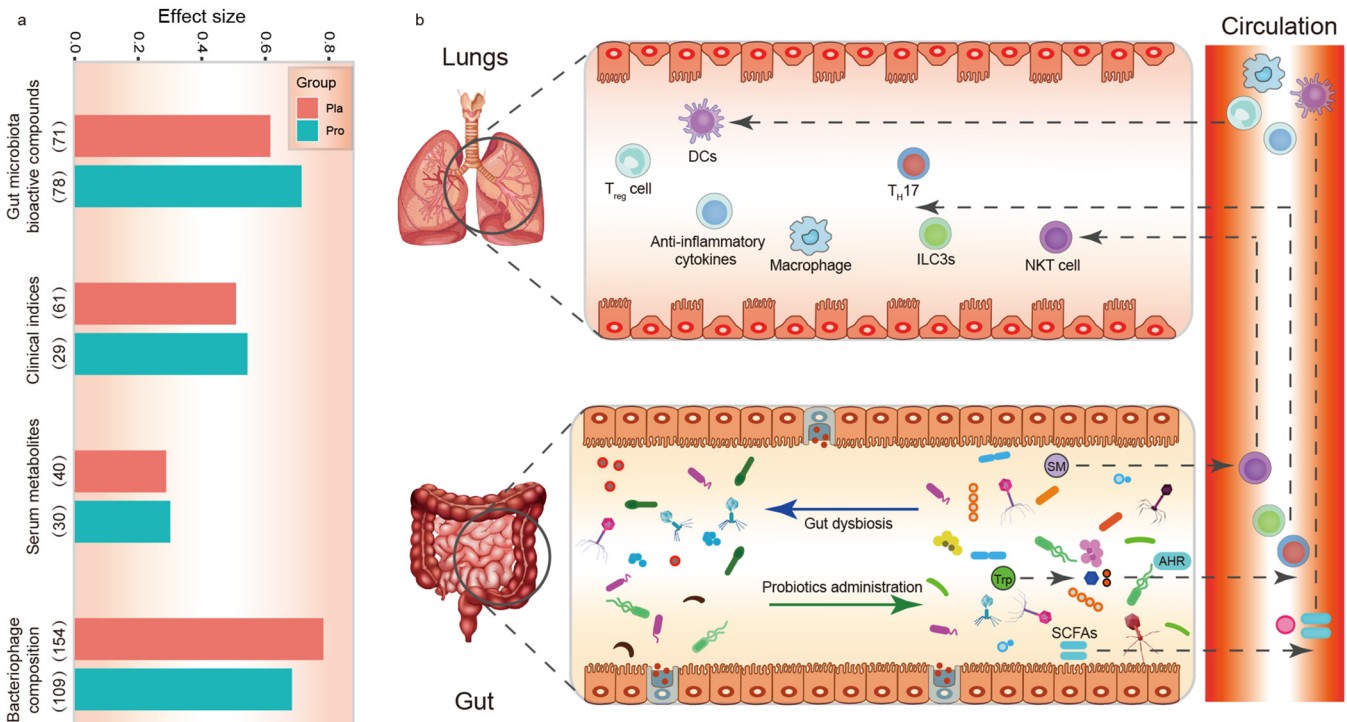

**FIG 6** Effect size of gut microbiota on the variation in the monitored features and proposed model of probiotic-driven pathways modulating the gut-lung axis in asthmatic patients. (a) The effect size of gut microbiota on the monitored features in the probiotic (Pro) and placebo (Pla) groups, including predicted gut bioactive compounds, clinical indexes, serum metabolomes, and virome. The number in parentheses represents the number of gut species-level genome bins (SGBs) that have an impact on the variation in the monitored features. (b) Schematic diagram illustrating key probiotic-driven pathways that modulated the gut-lung axis and host response. DCs, $T_{reg}$ cell, $T_H17$, ILC3s, and NKT cell represent dendritic cells, regulatory T cell, T helper 17, group 3 innate lymphoid cells, and natural killer T cells, respectively. SM, Trp, AHR, and SCFAs represent sphingomyelin, tryptophan, airway hyperresponsiveness, and short-chain fatty acids, respectively.

that, compared with the conventional drug treatment alone, co-administering probiotics with Symbicort Turbuhaler exerted stronger effects on the clinical symptoms, serum metabolome, and gut microbial metabolic potential but smaller effects on the gut virome.

## DISCUSSION

Probiotic adjuvant therapy has shown clinical efficacy in alleviating a number of medical conditions, including irritable bowel syndrome (28), anxiety symptoms (29), atopic dermatitis (30), and allergic asthmatics (21). Gut dysbiosis is a known trigger in multifactorial disorders, and the modulation of gut microbiota by probiotics is one of the potential mechanisms of the adjunctive efficacy (31). The gut microbiota is linked to human health and host immunity. Asthma is a multifactorial disease that is closely related to excessive lung inflammation, and the recently developed concept of gut-lung axis implicates bidirectional interactions and regulations between the two body compartments (4). Thus, this study investigated the adjuvant effect of a novel probiotic, Probio-M8, in managing asthmatic symptoms. Our results showed that, compared with those of the placebo group, co-administering Probio-M8 with conventional treatment significantly increased the ACT score, reduced the alveolar nitric oxide concentration, and lowered the fractional exhaled nitric oxide during/after the course of intervention, suggesting significant adjunctive efficacy of Probio-M8 co-administration with Symbicort Turbuhaler. The serum IgE levels were significantly and continuously increased in placebo but not Probio-M8 group during the trial period. It is known that an increased level of IgE could contribute to aggravation of symptoms in patients with bronchial asthma, although several previous clinical studies observed no significant change in the IgE level even with desirable clinical outcomes after probiotic or other

intervention (32–34). These discordant findings are suggestive of the existence of multiple disease pathways that could be differentially targeted by different therapeutics or management approaches.

A previous meta-analysis investigated the adjuvant clinical efficacy of probiotics in alleviating asthmatic symptoms (19). The study found that some probiotic strains (e.g., *Lactobacillus rhamnosus* GG) were effective in improving patients' symptoms, while no obvious beneficial effect was observed in other trials (e.g., *Lactobacillus rhamnosus* HN001) (35). The results obtained in human clinical trials vary greatly between studies, which could be related to different factors, particularly the study design. It is also known that the beneficial effects conferred by probiotics are strain specific. Thus, one valuable aspect of our current results is that the current results supported the clinical efficacy of Probio-M8 in attenuating asthmatic symptoms. Probio-M8 genomes were consistently detected in the fecal microbiome of probiotic receivers, suggesting that the bacteria transited through the host gastrointestinal tract successfully after ingestion; however, whether the probiotics could colonize or even propagate in the host gut would require further investigation. Our study hypothesized that the symptom improvement in the participants was related with modulation of subjects' gut microbiome and metabolism. Thus, in-depth fecal metagenomics and serum metabolomics analyses were performed to decipher significant probiotic-induced modulations in the subjects during the course of intervention.

First, it was observed that taking the routine regimen alone, but not co-administering together with Probio-M8, significantly reduced the alpha diversity of the gut bacterial microbiota and virome. Generally, the decrease in gut microbiota diversity is considered a less healthy state, as it indicates a disruption of gut homeostasis, and such disruption is particularly obvious when external agents, such as antibiotics and drugs, are applied, causing a drastic shift in the gut microbiota diversity and composition (36). Such disturbances might perturb normal energy metabolism and immunity (37) and even trigger onset or increase the severity of certain medical conditions (e.g., acute vascular rejection [38] and colitis [39]). Bacteria and viruses are the most extensively studied microbial communities related to asthmatic pathogenesis. Dickson et al. found that a decrease in bacterial diversity due to treatment was often associated with important clinical features of chronic lung diseases, including asthma (40). Moreover, a previous decrease in gut microbiome diversity predisposes children to the development of allergic asthma (41). Our results also showed that the inhalation of Symbicort Turbuhaler reduced the gut microbiota diversity and the combined intake of Probio-M8 could increase the resilience of the gut microbiome and maintain the baseline gut microbiota diversity level in the probiotic recipients.

On the other hand, the results of PCoA indicated that the symptom alleviation effect was not due to drastic shift in the structure of the gut microbiota and virome. Further analysis found that the probiotic-driven gut microbiota response was accompanied by significant changes in the prevalence of some SGBs. Our results showed that some potentially beneficial and anti-inflammatory SGBs, e.g., *Bifidobacterium animalis*, *Bifidobacterium longum*, *Prevotella* sp. CAG, *Roseburia hominis*, and *Ruminococcus callidus*, significantly increased after the intervention, while the SGBs representing *Clostridiales bacterium* and *Parabacteroides distasonis* decreased significantly. Other studies have also found that specific gut bacteria were closely related to asthma. For instance, more gut *Clostridium* and *Eggerthella lenta* were detected in asthmatic patients than in healthy people (6). Another study found more fecal *Streptococcus* and *Bacteroides* and less *Bifidobacterium* and *Ruminococcus* in children suffering from asthma (42). Furthermore, administering *Bifidobacterium animalis* reduced symptoms of viral respiratory infections in children (43). A 7-day randomized, double-blind, three-way crossover trial found that ingesting inulin or inulin with mixed probiotics, including *Bifidobacterium animalis* subsp. *lactis* Bb-12, improved airway inflammation and asthma control and modulated subjects' gut microbiome composition (44). Prophylactic supplementation of *Bifidobacterium longum* increased the fecal acetate level of mice and alleviated allergic airway inflammation/

airway hyperresponsiveness caused by T lymphocytes (45, 46). *Roseburia hominis* and *Ruminococcus callidus* are important butyrate-producers which increased the cecum butyrate level significantly and subsequently reduced airway hyperresponsiveness and the peripheral eosinophil count (47). These results suggested that the maintenance of colonic health and homeostasis relies on not only a diverse gut microbiota but also certain functional gut species or strains.

Our results also showed that co-administering probiotics enhanced anti-inflammatory microbial bioactive potential, metabolic pathways, and serum metabolomes. Probiotic treatment significantly increased the diversity of SGBs that possessed pathways related with butyrate synthesis and tryptophan metabolism. Interestingly, co-administering Probio-M8 with the conventional drug increased serum tryptophan levels. Short-chain fatty acids, including butyrate, propionate, and acetate, are produced by a wide variety of intestinal microbes through fermentation of dietary fiber, which might help inhibit proinflammatory responses in the lungs (48). Feeding mice with SCFAs during pregnancy and weaning could protect the offspring from allergic lung inflammation, and butyrate effectively induced the production of T regulatory cells in the lungs of offspring (49). Multiple human observational studies reported that reductions in fecal SCFA during infancy were associated with development of asthma later in life (42, 50). On the contrary, children with large amounts of fecal butyrate and propionate showed significantly less atopic sensitization (49). Collectively, these findings supported that SCFAs could protect against the development of asthma. The underlying mechanism could be that some SCFAs, especially butyrate, serve as energy source for colonocytes, while unmetabolized SCFAs enter the peripheral circulation and distal body sites (such as the lungs), where they modulate the activity of T regulatory lymphocytes and various regulatory cytokines (Fig. 6b) (51, 52). Gut microbes are the major participants in tryptophan metabolism, which protects the host from developing asthma through indoleamine 2,3-dioxygenase-1 pathway and aryl hydrocarbon receptor activation (53). In murine models of asthma, activation of aryl hydrocarbon receptors reduced airway inflammation and hyperresponsiveness (54). Tryptophan metabolites interact with the aryl hydrocarbon receptor to promote immune homeostasis, enhance Th17 and T regulatory cell differentiation, maintain homeostasis of innate lymphocytes, and exert anti-inflammatory and antioxidant effects in the systemic circulation (53, 55).

In addition, probiotic intake increased the predicted levels of linoleoyl ethanolamide, $C_{18:1}$, cholesterol ester, and $C_{18:0}$ sphingomyelin, as well as the serum sphingomyelin level. Linoleoyl ethanolamide is a bioactive fatty acid ethanolamide that exerts anti-inflammatory effects by inhibiting NF-$\kappa$B signaling (56). Sphingomyelin is a sphingolipid found in animal cell membranes (57). A previous study found that, compared with that of age-matched control children, asthmatic patients had a significantly lower level of sphingomyelin in the whole blood (58). In addition, two human studies found that low levels of fecal sphingolipids in early life were related to food allergies (59, 60). Thus, sphingolipid metabolites might protect against asthma, but further research would still be needed to confirm the biochemical pathways and mechanisms of attenuation of the pathophysiology in asthma.

Notably, the current study found that changes in the alpha diversity of the gut virome correlated strongly with the gut microbiome during the course of treatment, suggesting that the gut virome was likely comodulated in a way similar to that of the gut microbiota. Likewise, co-administering probiotics helped maintain the alpha diversity of bacteriophage, which dropped significantly in the placebo group. The gut virome has recently been shown to be closely linked with human health via surprising mechanisms. For example, bacteriophages and their components directly stimulated mammalian immune responses (61) and shaped the gut microbiota (62). The current findings implicate a tripartite relationship among the host gut bacteria, bacteriophages, and immune system.

In conclusion, this study showed that Probio-M8 was effective in alleviating asthmatic symptoms. Co-administering probiotics with routine drug treatment maintained

the alpha diversity and stability of the gut bacterial microbiota and virome in asthmatic patients. Meanwhile, the application of probiotics has brought about health-promoting effects to asthmatic patients by increasing certain anti-inflammatory serum metabolites, gut bioactive metabolites, and related pathways. These changes directly modulated the gut-lung axis and enhanced the treatment efficacy. Our results demonstrated that co-administering Probio-M8 synergized with conventional therapy to alleviate diseases associated with the gut-lung axis, like asthma, expanding the treatment options of related chronic disorders.

## MATERIALS AND METHODS

**Experimental design and subject recruitment.** This work performed a 3-month longitudinal study. A total of 58 asthmatic patients were recruited from the Medical Clinic of Weihai Municipal Hospital (Weihai City, Shandong Province, China). Patients were screened by the inclusion and exclusion criteria. Inclusion criteria: male or female, 18 to 75 years old, met the diagnostic criteria for asthma, with temporarily stable asthmatic symptoms in nonacute attack stage that could be controlled by drug treatment. All included patients expressed willingness to commit throughout the trial. Exclusion criteria: subjects with any history of major disease, such as gastrointestinal diseases, mental illness, or type I diabetes, long-term requirement for glucocorticoids or other immunosuppressive agents, antibiotics intake 1 month before the trial, intake of probiotics-based products 1 month before the trial started. Subjects that did not complete the trial for any reason were excluded.

The first round of screening excluded one patient because of the subject's unwillingness to participate in the trial. The remaining 57 patients were randomly assigned to probiotic ($n = 30$) and placebo ($n = 27$) groups. After being informed of the specific experimental guidelines and details, one patient from each group withdrew from the trial, and 29 and 26 subjects remained in the probiotic and placebo groups subsequently. No significant difference was found in age and sex between Probio-M8 and placebo groups ($P > 0.05$; Table S1). Patients were not informed about the group allocation throughout the trial.

All patients were encouraged to eat three regular meals per day at fixed times. At the same time, patients were asked to avoid ingesting any other probiotic products or antibiotics. These patients received medication for 2 to 3 months after being hospitalized, and when their medical conditions became relatively stable as assessed by our collaborative clinicians, they were randomized into designated groups according to the planned setup of the current trial. The probiotic group received both the Symbicort Turbuhaler and one sachet of Probio-M8 powder per day (2 g per sachet; $3 \times 10^{10}$ CFU/sachet/day), while the placebo group received the Symbicort Turbuhaler and placebo material. The intervention continued for 3 months. The Probio-M8 powder and placebo material (one sachet per day) were manufactured by Jinhua Yinhe Biological Technology Co. Ltd., China. The sachets of both the probiotic and the placebo materials appeared as light pink powder and were identical in weight, taste, and appearance.

The patients' asthma control indicators, including ACT score, lung function indices (peak expiratory flow [PEF], forced expiratory volume in 1 s [FEV1], forced vital capacity [FVC]), fractional exhaled nitric oxide (FeNO), alveolar nitric oxide concentration (CANO), peripheral eosinophil counts, and IgE, were recorded at days 0, 30, and 90. Fecal and blood samples were also collected at days 0, 30, and 90. However, due to the withdrawal of patients and the failure of continuous donation of fecal and blood samples at all three time points, only 17 and 14 patients remained in the probiotic and placebo groups, respectively (Fig. S1a). Fresh blood samples were taken to measure eosinophil count, and then all samples were stored at −80°C until analysis.

**DNA extraction and shotgun metagenomic sequencing.** DNA was extracted from fecal samples with the QIAamp fast DNA stool minikit (Qiagen, Hilden, Germany). Shotgun metagenomic sequencing was performed on all samples with an Illumina HiSeq 2500 instrument. Libraries were constructed using NEBNext Ultra DNA library prep kit for Illumina (NEB, USA) following the manufacturer's recommendations to generate DNA fragments of ~300 bp; paired-end reads were generated by sequencing 150 bp in the forward and reverse directions.

A total of 93 stool samples were shotgun sequenced ($n$ of 17 and 14 for Probio-M8 and placebo groups; sampling at days 0, 30, and 90 for each individual), generating 630.62 Gbp of high-quality paired-end reads (6.78 ± 0.97 Gbp/sample; range = 4.45 to 9.69 Gbp) for downstream analysis. Low-quality (sequences shorter than 60 nucleotides [nt]) and host-contaminated reads were filtered through the KneadData quality control pipeline (http://huttenhower.sph.harvard.edu/kneaddata).

**Reads assembly, contig binning, genome dereplication.** Reads of each sample were assembled into contigs using MEGAHIT (63), with an average $N_{50}$ length of 21.50 kbp (Table S2). Contigs greater than 2,000 bp were selected for binning using MaxBin2 (64), MetaBAT2 (65), and CONCOCT (66) with default options, generating 5,147, 7,319, and 7,881 raw bins from the initial contigs, respectively. Then, the results of the three binners were combined to obtain 3,762 metagenome-assembled genomes (MAGs) using MetaWRAP's bin refinement module (67). Sample reads were mapped back to the corresponding contigs using BWA-MEM (68), and the read depth was calculated using Samtools (69) and the jgi_summarize_bam_contig_depths function in MetaBAT2.

The levels of completeness and contamination of MAGs were evaluated by CheckM (70), and the MAGs were classified as high quality (completeness $\geq$ 80%, contamination $\leq$ 5%), medium quality (completeness $\geq$ 70%, contamination $\leq$ 10%), and partial quality (completeness $\geq$ 50%, contamination $\leq$ 5%) (29). The high-quality genomes were clustered, and the most representative genome from each replicate set

was selected by dRep (71) with the parameter settings -pa 0.95 and -sa 0.95. Finally, 389 species-level genome bins (SGBs) were extracted from the pool of single representative genomes.

**Taxonomic annotation of SGBs and their relative abundance.** The SGBs were annotated through the Kraken2 and NCBI Nonredundant Nucleotide Sequence Database (retrieved in November 2020), and the predicted genes were searched against the UniProt Knowledgebase (UniProtKB, release 2020.11) using the blastp function of DIAMON with default options. A total of 389 SGBs were cross-compared with the IGG data set (https://github.com/snayfach/IGGdb; contained 16,136 nonredundant representative human gut genomes) and the MAGs data set (contained 154,723 nonredundant genomes, which were mainly gut sample data sets of the global population) to evaluate the novelty of SGBs of the current data set. The sample reads were mapped to contigs through BBMap (https://github.com/BioInfoTools/BBMap) using the parameter "minid = 0.95 idfilter = 0.95." The sam file was used as the input file, and the pileup.sh tool was used to calculate the coverage of contigs. The average content of SGBs in each contig was calculated and expressed in reads per kilobase per million (RPKM) by an in-house script.

**Prediction of gut metabolic modules and active metabolites.** For each SGB, the predicted open reading frames (ORFs) were compared to Kyoto Encyclopedia of Genes and Genomes (KEGG) Orthologies (KOs) database to predict the key metabolic modules. To narrow down the GMMs relevant to this study, first, GMMs described in Darzi et al. (72), Magnusdottir et al. (73), and the MetaCyc metabolic database (74) were retrieved. Then, target modules, such as SCFAs, histamine, polyunsaturated fatty acids, bile acids, tryptophan, sphingolipids, vitamin D, and other metabolic modules, were extracted from 159 module databases after several key publications relating to the onset, development, pathophysiology, and immune responses of asthma were consulted (13, 51, 75, 76). The distribution of synthesis and/or degradation modules in the SGBs was determined by Omixer-RPM (72) using the parameter -c 0.66.

Then, the spectrum of gut active metabolic compounds was also predicted. Briefly, one million reads per sample were extracted and compared using seqtk (https://github.com/lh3/seqtk) and the blastx function of DIAMON with parameters –query-cover 90 and –id 50, respectively. The gene abundance in each sample was calculated through the best hit of each gene. Finally, the MelonnPan-predict pipeline (77) was used to convert gene abundance into predicted active metabolites profiles.

**Analysis of serum metabolome by LC-MS.** Serum metabolites were extracted (78). Briefly, serum samples were thawed at 4°C, and 200 $\mu$l of each sample was transferred to a 2 ml sterile centrifuge tube and vortex mixed with 800 $\mu$l of methanol-water solution (4:1, vol/vol) for 1 min. The mixture was allowed to stand at $-20$°C for 60 min before centrifugation at 17,000 $\times$ $g$ for 15 min at 4°C. Five hundred microliters of the supernatant was vacuum dried at room temperature. Then, the dried sample was dissolved in 200 $\mu$l of acetonitrile-water solution (1:1, vol/vol). The samples were then centrifuged at 12,000 $\times$ $g$ for 10 min at 4°C, and the supernatants were passed through 0.22-$\mu$m microporous membrane filter before further analysis.

The MS analysis of serum samples was performed with an Agilent 6545A QTOF (Agilent Technologies, Santa Clara, CA) in both positive and negative ion modes. The quality control (QC) sample was prepared by mixing equal amounts (10 $\mu$l) of each sample, and the QC sample was injected five times to evaluate the stability of the system before actual sample analysis. The raw data were filtered by Progenesis QI platform and then preprocessed using MetaboAnalyst (https://www.metaboanalyst.ca/), and features with more than 50% missing values were excluded. Noninformative features, such as baseline noise and near-constant values, were excluded by using mean, median, and standard deviation, and the low repeatability features (evaluated based on comparison with the QC sample) with a peak area relative standard deviation (RSD) of >20% was also excluded from the analysis.

The metabolomic data were analyzed by partial least-squares-discriminant analysis (PLS-DA), and the variable importance in projection (VIP) score was used for screening differential abundant metabolites using the SIMCA-P +14.0 software (Umetrics, MKS Instruments Inc., Sweden). Differential abundant biomarkers were manually inspected based on peak shape and signal-to-noise (S/N) ratio, and their tandem mass spectrometry (MS/MS) spectra were cross-compared with the Blood Exposome Database (https://bloodexposome.org) to identify the features of interest.

**Taxonomic annotation of virome and their relative abundance.** After assembling by MEGAHIT, contigs greater than 1,000 bp were selected for further identification of potential viral features through VIBRANT and DeepVirFinder. The results recovered from these two tools were combined using in-house scripts, which returned 219,729 potential DNA virus sequences. The virus contigs were pairwise blasted, and highly consistent viruses with 95% nucleotide identity and 80% sequence coverage were further clustered to obtain 31,441 DNA viral operational taxonomic units (vOTUs) using CD-HIT. CheckV classified the vOTUs into different groups based on assembling quality and completeness, resulting in vOTUs of complete quality (330), high quality (1,216), medium quality (1,355), low quality (10,177), and undetermined quality (18,363). All except the undetermined-quality vOTUs were further analyzed (Fig. S1b).

The vOTUs were annotated with the latest NCBI virus database using the blastp function of DIAMOND with options –query-cover 70 and –id 30. If more than one-third of the annotated proteins of a vOTU belonged to the same family, the vOTU would be assigned to that viral family (79). The average abundance of vOTUs was calculated by the CoverM-contig pipeline (https://github.com/wwood/CoverM) with the following parameter settings: –min-read-percent-identity 0.95, –min-read-aligned-percent 0.5, –proper-pairs-only, and –exclude-supplementary.

**Statistical analyses.** All statistical analyses were performed using the R software (v.4.0.2). We calculated the species diversity through the R package (vegan and optparse) based on the normalized RPKM abundance. Principal coordinates analysis (PCoA) and PLS-DA analysis were performed and visualized

using the R package vegan, mixOmics, and ggpubr. The Adonis *P* value was generated based on 999 permutations. Kruskal-Wallis test, Wilcoxon test, and *t* test were used to evaluate differences in various variables between groups; *P* values were corrected for multiple testing using the Benjamini-Hochberg procedure. The method of effect size analysis was performed according to Yan et al. (79). The centered log-ratio (clr) abundance transformation was performed using the R package microbiome. Procrustes analysis in the vegan package was used to determine similarity between two multivariate axes (permutations = 999). All graphical presentations were generated under the R and Adobe Illustrator environment.

**Data and code availability.** Sequencing data and analysis codes are available in NCBI-SRA (BioProject: PRJNA722129; https://www.ncbi.nlm.nih.gov/bioproject/PRJNA722129/) and under https://github.com/TengMa-Cleap/Probiotics-relieve-human-asthma-project/. Metabolite data are also provided to the Mass Spectrometry Interactive Virtual Environment (MSV000088033; https://massive.ucsd.edu/ProteoSAFe/dataset.jsp?task=e185c780a113402b8b71b886223f1a22).

## SUPPLEMENTAL MATERIAL

Supplemental material is available online only.

**SUPPLEMENTAL FILE 1**, PDF file, 1.4 MB.

## ACKNOWLEDGMENTS

This research was supported by the National Natural Science Foundation of China (grant numbers 31720103911 and 31972083), Inner Mongolia Science & Technology Major Projects (grant number ZDZX2018018), and Shandong Science & Technology Development Project of Medical and Health (grant number 202003020112).

The study was approved by the Ethics Committee of the Weihai Municipal Hospital (project number 201816) and was registered on the Chinese Clinical Trial Registry (http://www.chictr.org.cn/; registration number ChiCTR1800017162). Informed consent was obtained from all recruited subjects prior to the study.

We declare no conflicts of interest.

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
