## [Reviewer comments · Microbiology Spectrum]

Microbiology Spectrum

Adjunctive probiotics alleviates asthmatic symptoms via modulating the gut microbiome and serum metabolome

Ailing Liu, Teng Ma, Ning Xu, Hao Jin, feiyan zhao, Lai-Yu Kwok, Heping Zhang, Shukun Zhang, and zhihong sun

Corresponding Author(s): zhihong sun, Inner Mongolia Agricultural University

Review Timeline:

Submission Date:	July 13, 2021
Editorial Decision:	July 30, 2021
Revision Received:	August 17, 2021
Editorial Decision:	August 26, 2021
Revision Received:	August 30, 2021
Accepted:	August 31, 2021

Editor: Wei-Hua Chen

Reviewer(s): Disclosure of reviewer identity is with reference to reviewer comments included in decision letter(s). The following individuals involved in review of your submission have agreed to reveal their identity: Bong-Soo Kim (Reviewer #1); Mehmet Demirci (Reviewer #2)

Transaction Report:

DOI: <https://doi.org/10.1128/Spectrum.00859-21>

July 30, 2021

Dr. Teng Ma
Inner Mongolia Agricultural University
Hohhot
China

Re: Spectrum00859-21 (Adjunctive probiotics alleviates asthmatic symptoms via modulating the gut metagenome and serum metabolome)

Dear Dr. Teng Ma:

Thank you for submitting your manuscript to Microbiology Spectrum. Your manuscript now has been seen by two reviewers. As you will see below, while they find your work of potential interest, they have raised quite substantial concerns that must be addressed. In light of these comments, we cannot accept the manuscript for publication, but would be interested in considering a revised version that addresses these serious concerns.

We hope you will find the reviewers' comments useful as you decide how to proceed. Should further experimental data or analysis allow you to address these criticisms, we would be happy to look at a substantially revised manuscript. However, please bear in mind that we will be reluctant to approach the reviewers again in the absence of major revisions.

When submitting the revised version of your paper, please provide (1) point-by-point responses to the issues raised by the reviewers as file type "Response to Reviewers," not in your cover letter, and (2) a PDF file that indicates the changes from the original submission (by highlighting or underlining the changes) as file type "Marked Up Manuscript - For Review Only". Please use this link to submit your revised manuscript - we strongly recommend that you submit your paper within the next 60 days or reach out to me. Detailed information on submitting your revised paper are below.

Link Not Available

Sincerely,

Wei-Hua Chen

Journals Department
Reviewer comments:

Reviewer #1 (Comments for the Author):

This study analyzed the effects of probiotics on asthmatic symptoms by modulating gut microbiome and serum metabolites. The topic is interesting and positive effects of co-administration of probiotics and conventional therapy. However, I have several major concerns about the way the analyses were conducted and interpretation of the results.

1. Most results showed the comparison of longitudinal changes within the same group. However, authors described their results with comparison of results between probiotics and placebo groups. If author showed the differences and comparison between two groups, the changed values or detected values at each time point should compare directly between groups with statistical significance. For example, Fig 1a, Fig 2a,b Fig2c,d etc. should directly show the difference of score between probiotics and placebo groups.
2. Final analyzed subject number was 31. Author should clearly indicate this in abstract.
3. Age ranges of studied subjects were too broad (18-75 years old). The gut microbiome shifts according to ages. In particular, they are different between adults and elderly as reported several studies. Authors should showed the independency of their results from age factor.
4. For Table S1, authors should show final analyzed subjects used in results of present study. In addition, clinical features should summary and compare between groups with statistic significance.
5. Asthma symptom and their associated gut microbiome or metabolome could be different by severity. Why did not author consider the severity of asthma?
6. Please use FDR corrected p value as possible.
7. Why did not consider cytokine or immunological assay in this study? Asthma is an immune-mediated disease, and the gut microbiome can be related to systemic immune features.
8. Difference of each detected features between two groups was already detected at 0 day. Authors should normalize or compare changed values between groups. In particular, proportion of bacterial species, predicted metabolites, and serum metabolites were different between two groups at 0 day.
9. Fig 3a should modify to understand. Present form can not clearly show differences between groups. In addition, procrustes analyses figures was difficult to read and understand.
10. Authors described the Probio-M8 could colonize and propagate in the host gut by detection of Probio-M8 strain sequences in the samples. How can we conclude the 'colonization' by detecting sequence in samples? The ingested strain can simply be passed through the digestive tract and it

can be detected in the feces.

Minor concerns

modulating the gut metagenome -> modulating gut microbiome in title and whole manuscript

Results in abstract should described the direct comparison between groups.

line 142: Exclusion criteria: no history of major disease..-> Were subjects with no history of major disease excluded in this study? I think that you wanted to exclude subjects with any history of major disease.

line 156: relative stable intake of protein and dietary fiber...-> How can we determine relative stable intake? Do you have any other criteria to determine?

line 177: Fecal DNA -> DNA was extracted from fecal samples...

line 179: Which library kit did you use? Please clarify.

line 201-202: Why the contamination values were higher in medium quality than in partial quality? Please show the supporting scientific data to determine these criteria.

In method, there were no description about the calculation of diversity. The comparison of diversity between samples should conducted after normalization of read number.

line 376-379. Please provide criteria to determine focusing GMM.

Reviewer #2 (Comments for the Author):

The manuscript is interesting in that it provides important data on asthma treatment and probiotic supplementation. I think that the following missing points should be added to the article.

It is necessary to create a new summary table (by creating mean, max, min, SD) of both the demographic data and the Asthma symptoms control indices data (Table S1) of the patients who received probiotic treatment and were included in the placebo group. These data should be compared statistically and it should be shown that there is no difference between the groups. In addition, other data that may affect the daily life of patients such as body mass index, smoking and alcohol consumption should be added to this table and compared between groups, and if there is a difference, additions should be made considering the effect of these on the data.

How many of the patients had a primary diagnosis or how many had been receiving treatment for how long, The absence of any data on this is a shortcoming.

Although the blood of the patients was taken, IgE levels were not detected?

Staff Comments:

Preparing Revision Guidelines

For complete guidelines on revision requirements, please see the Instructions to Authors at [link to page]. **Submissions of a paper that does not conform to Microbiology Spectrum guidelines will delay acceptance of your manuscript.**

Please return the manuscript within 60 days; if you cannot complete the modification within this time period, please contact me. If you do not wish to modify the manuscript and prefer to submit it to another journal, please notify me of your decision immediately so that the manuscript may be formally withdrawn from consideration by Microbiology Spectrum.

If you would like to submit an image for consideration as the Featured Image for an issue, please contact Spectrum staff.

15 August 2021

Dear Editor:

Thank you for your and the reviewers' comments and suggestions on our manuscript. The comments and suggestions are valuable for improving our manuscript. We have read the comments carefully and revised accordingly, the revised portion of the manuscript is shown in red.

We hope our revised version will now be acceptable for publication in Microbiology Spectrum, and we look forward to hearing from you. Thank you for your time and consideration.

Best regards,

Teng Ma

Answers to reviewers:

Reviewer #1:

[1] Most results showed the comparison of longitudinal changes within the same group. However, authors described their results with comparison of results between probiotics and placebo groups. If author showed the differences and comparison between two groups, the changed values or detected values at each time point should compare directly between groups with statistical significance. For example, Fig 1a, Fig 2a,b Fig2c,d etc. should directly show the difference of score between probiotics and placebo groups.

Answer: Thank you very much for your comment. We very much agree with you. Direct horizontal comparisons of clinical features, gut microbiota, serum metabolome, and other data between probiotics and placebo groups are now shown in the manuscript to illustrate the clinical efficacy of probiotics. The Results section, as well as relevant tables and figures, are updated accordingly. Please see Figure 1a, Figure 2a, 2b, Figure 4a, Figure 5b, 5c, and Table S3.

[2] Final analyzed subject number was 31. Author should clearly indicate this in abstract.

Answer: Thank you very much for your comment. We have added a brief but clear description of participant recruitment with indication of the final analyzed subject number of 31 in the Abstract. The number of finally analyzed set of samples (31) is also clearly indicated in the Methods section. The details of study design and participant recruitment process are shown in Figure S1a. Please see line 29-35.

[3] Age ranges of studied subjects were too broad (18-75 years old). The gut microbiome shifts according to ages. In particular, they are different between adults and elderly as reported several studies. Authors should showed the independency of their results from age factor.

Answer: Thank you very much for your comment. We very much agree with your opinion that the age factor does have an important impact on the gut microbiota. The

age range of initially recruited subjects was 18-75 years old; however, since some subjects dropped out, the actual age range of the 55 qualified asthma patients was 28-72 years old (mean=55.78; median=58; 74.54% of them were between 50 and 70-year-old). The table below shows the demography data of the subjects, which is now included as part of Table S1. Statistical analysis found no significant difference in the age distribution between the two groups ($P<0.05$). In addition, there was also no significant difference in other factors, including sex ratio, BMI, habit of alcohol consumption, and history of smoking. Such information is now described in the Results section in the updated manuscript (please see line 371-374).

	Probiotics_group	Placebo_group	P_value
Male	12	11	-
Female	17	15	-
Age	54.62±9.61	57.08±10.46	0.17
BMI	24.41±2.66	25.11±3.82	0.7
Alcohol consumption	28.28±30.25	20.38±27.20	0.39
History of smoking	21.33±9.07	29.00±12.36	0.3

[4] For Table S1, authors should show final analyzed subjects used in results of present study. In addition, clinical features should summary and compare between groups with statistical significance.

Answer: Thank you very much for your comment. Your suggestion is very helpful for improving the quality of our manuscript, and we have revised the table. The details of subject recruitment and sample collection have been updated in Table S1. In addition, monitored clinical features (e.g., asthma control test score, alveolar nitric oxide concentration, fractional exhaled nitric oxide, IgE level, eosinophil counts...) were compared between groups in Table S3.

[5] Asthma symptom and their associated gut microbiome or metabolome could be different by severity. Why did not author consider the severity of asthma?

Answer: Thank you very much for your comment. We very much agree with your opinion that the severity of the disease could be associated with patients' gut

microbiome and serum metabolites in asthmatic patients. Indeed, the original thought of study design was to recruit asthmatic patients of four severity levels (mild, moderate, severe, and critical) for investigating effects of probiotic intervention on clinical outcomes in different groups. However, after consulting and discussing with our collaborative clinician partners, we agreed that it would be better to start with a smaller scale trial presented in this work, as it would be challenging and might be over-ambitious to cover patients of different severity due to problems like compliance of patients in medical treatment and probiotic intervention, difficulties in follow-up of patients' conditions, symptom control and so on. Moreover, asthma is a serious medical condition, and, ethically, one prime concern in our study design was to ensure patients' health and safety. Thus, we decided only to recruit patients with stable and manageable asthmatic symptoms in non-acute attack stage in this initial study, as well as keeping a relatively small cohort of subjects to ensure every participant was well taken care of. In future, this work will be elaborated to cover asthmatic patients of different severity. We hope and do believe (particularly based on the support of the current results) that applying probiotic as an adjunctive treatment would be beneficial to asthmatic patients. Thank you again for your comments and suggestions.

[6] Please use FDR corrected p value as possible.

Answer: Thank you very much for your comment. In the revised manuscript, we have applied FDR corrected p value wherever possible. Please see Table S3, S6, S8 and S9.

[7] Why did not consider cytokine or immunological assay in this study? Asthma is an immune-mediated disease, and the gut microbiome can be related to systemic immune features.

Answer: Thank you very much for your comment. Both reviewers mentioned this important issue. Indeed, we did conduct immunological assays to determine the serum levels of IgG, IgM, IgA, C-reactive protein, and IgE, but data were not included in our first manuscript due to non-significant differences in most of these parameters. IgE is

the most relevant immunoglobulin in patients with bronchial asthma, resulting in the aggravation of asthma symptoms. Although the serum IgE levels in the placebo group increased significantly and continuously during the course of study, no significant difference was found between probiotics and placebo groups. In fact, several previous clinical studies also found no significant change in serum IgE level after probiotics or other intervention even with a good clinical efficacy (Ou et al., 2012; Joks et al., 2005; Cao et al., 2018). These findings are suggestive of the existence of multiple disease pathways, which could be differentially targeted by different therapeutics or management approaches. We have updated the manuscript in the Results and Discussion sections. Please see line 338-340 and 516-523.

[8] Difference of each detected features between two groups was already detected at 0 day. Authors should normalize or compare changed values between groups. In particular, proportion of bacterial species, predicted metabolites, and serum metabolites were different between two groups at 0 day.

Answer: Thank you very much for your comment. The Reviewer is right that there were differences in some of the monitored features at the baseline level (0 day) between the Probio-M8 and placebo groups. On one hand, it is important to compare between groups. But, on the other hand, we think that it would be important to identify differences in response to treatment between the two groups during/after the intervention, as well as between the same individual during and after intervention compared with baseline. This would be important in accurately identifying key species or metabolites specifically regulated by probiotic intervention. Our manuscript aimed to address differences in both directions.

To address the Reviewer's concern, the related figures have been modified to clarify the results. In the updated manuscript, Figure 2c (Baseline: no significance; Trial period: changed significantly between the 2 groups), Figure 3c, and Figure 4c show the changes in species, predicted metabolites, and serum metabolites between the Probio-M8 and placebo groups during/after the intervention, respectively.

[9] Fig 3a should modify to understand. Present form cannot clearly show differences between groups. In addition, procrustes analyses figures was difficult to read and understand.

Answer: Thank you very much for your comment. Figure 3a showed the significantly differential species-level genome bins (SGBs) that encoded relevant gut metabolic modules between the Probio-M8 and the placebo groups at different time points. We revised the annotation information of the horizontal SGBs and indicated significant differences in SGBs in the figure.

Procrustes analysis is a statistical technique that utilizes data dimensionality reduction methods (such as PCoA, NMDS, and CCA), to display multi-omics datasets in low-dimensional space to evaluate the similarities and differences between datasets. In recent years, it has been increasingly used to evaluate the relationship between the microbiome/metabolome/phenotype datasets (Ashrafi et al., 2020; Karl et al., 2017; Mchardy et al., 2013). We have expanded the principles of the analysis and the meaning of our results in the updated manuscript. We hope that the information improves the readability. Please see line 413-418.

[10] Authors described the Probio-M8 could colonize and propagate in the host gut by detection of Probio-M8 strain sequences in the samples. How can we conclude the 'colonization' by detecting sequence in samples? The ingested strain can simply be passed through the digestive tract and it can be detected in the feces.

Answer: Thank you very much for your comment. We agree with the Reviewer. Thus, we have modified the description as: "...suggesting that the ingested Probio-M8 strain could easily pass through the digestive tract.". Please see line 361-362.

[11] modulating the gut metagenome -> modulating gut microbiome in title and whole manuscript.

Answer: Thank you very much for your comment. We have made correction in the title and throughout the manuscript as suggested. Please see the title and line 538.

[12] Results in abstract should described the direct comparison between groups.

Answer: Thank you very much for your comment. We have modified the abstract to describe the comparison between groups. Please see line 35-40.

[13] line 142: Exclusion criteria: no history of major disease..-> Were subjects with no history of major disease excluded in this study? I think that you wanted to exclude subjects with any history of major disease.

Answer: Thank you very much for your comment. Yes, I actually want to state "exclude subjects with any history of major disease". Thank you for your reminder, I have modified it. Please see line 146.

[14] line 156: relative stable intake of protein and dietary fiber...-> How can we determine relative stable intake? Do you have any other criteria to determine?

Answer: Thank you very much for your comment. These were general instructions given to participants prior to the trial, so there were no specific criteria or standards for diet control. At the beginning of the trial, we encouraged patients to eat three meals a day at a fixed time, to avoid irregular eating habits, to avoid partial and picky eating, and to have balanced nutrition. On the one hand, balanced and regular diet intake would be helpful for the treatment and rehabilitation; and on the other hand, it could help avoid the impact of drastic dietary changes on the intestinal microbiota and metabolites.

We thank and agree with the Reviewer's comment. The original description ("...relative stable intake of protein and dietary fiber...") in the manuscript could be misleading and was deleted. Please see line 159-160.

[15] line 177: Fecal DNA -> DNA was extracted from fecal samples...

Answer: Thank you very much for your comment. We have corrected it. Please see line 184-185.

[16] line 179: Which library kit did you use? Please clarify.

Answer: Thank you very much for your comment. We have clearly described the information of the library kit [NEBNext® Ultra™ DNA Library Prep Kit for Illumina (NEB, USA)] in the Methods section. Please see line 186-188.

[17] line 201-202: Why the contamination values were higher in medium quality than in partial quality? Please show the supporting scientific data to determine these criteria.

Answer: Thank you very much for your comment. Our literature search found that, generally speaking, the consensus for high-quality genomes was at least at a level of completeness $\geq 80\%$ and contamination $\leq 5\%$ (as in our study), but these threshold standards varied between studies. For example, Almeida et al. (2019) defined the medium quality of bins as completeness $\geq 50\%$ and contamination $< 10\%$, which was employed by another study (Xie et al., 2021). On the other hand, Parks et al., 2017 used the following thresholds: “near-complete genomes” (completeness $\geq 90\%$; contamination $\leq 5\%$), “medium-quality genomes” (completeness $\geq 70\%$; contamination $\leq 10\%$), and “partial genomes” (completeness $\geq 50\%$; contamination $\leq 4\%$).

It is common to see that “contamination values were higher in medium quality than in partial quality”. Although “completeness” and “contamination” are two independent parameters indicating the genome assembling quality, there needs to be a balance between the two parameters to achieve an acceptable genome quality level. “Partial genome” is supposed to have the lowest quality among the three categories of genomes; however, it is still important to ensure a relatively high specificity. The specificity of the partial genome would be largely compromised if both the levels of “completeness” and “contamination” are simultaneously and largely relaxed.

Indeed, all follow-up analyses in our study only included high-quality genomes but not medium-/partial- genomes to ensure that inferences drawn in our study were derived from highly accurate and specific data of taxonomic and functional annotations from high-quality genomes.

[18] In method, there were no description about the calculation of diversity. The comparison of diversity between samples should be conducted after normalization of read number.

Answer: Thank you very much for your comment. We first used the BBDMap tool to calculate the species distribution, and the abundance of detected taxa was expressed in “Reads Per Kilobase Million (RPKM)”, which was calculated after normalization of read number. Then, we calculated the species diversity through the R package (vegan and optparse) based on the RPKM abundance. We have added the detail in the Methods section, and the analysis codes have been released under my github account (<https://github.com/TengMa-Cleap>). Please see line 304-306.

[19] line 376-379. Please provide criteria to determine focusing GMM.

Answer: Thank you very much for your comment. First, we used the gut metabolic modules (GMM) described in the two published literatures (Darzi et al., 2016; Magnúsdóttir et al., 2017; 159 modules in total) and MetaCyc metabolic database as reference. Then, based on several high-quality literatures (Lee-Sarwar et al., 2020; Depner et al., 2020; Platten et al., 2019; Carr et al., 2019) related to asthma development, pathophysiology, and immune system, target modules such as SCFAs, histamine, polyunsaturated fatty acids, bile acids, tryptophan, sphingolipids, vitamin D, and other metabolic modules, were extracted from 159 module databases. Finally, Omixer-RPM with the parameter -c 0.66 has been used to identify the metabolic modules involved in the SGBs. Thank you for your question. We have updated the manuscript methods and results. Please see line 236-244.

Reviewer #2:

[1] It is necessary to create a new summary table (by creating mean, max, min, SD) of both the demographic data and the Asthma symptoms control indices data (Table S1) of the patients who received probiotic treatment and were included in the placebo group. These data should be compared statistically and it should be shown that there

is no difference between the groups.

Answer: Thank you very much for your comment, your suggestion has helped improve our data presentation. In response, we have revised the Tables relevant to patients demographic data and asthmatic symptom indexes. In the updated manuscript, Table S1 showed the demographic data of 55 asthmatic patients included in this study, as well as comparison between the two groups with statistical analysis ($P>0.05$ in the analyzed factors). Table S3 and Figure 1 display differences in the clinical features between the Probio-M8 and placebo groups at each time point. In addition, we have illustrated the detail of participant recruitment flow in Figure S1. Please see Table S1, Table S3, Figure 1, and Figure S1.

[2] In addition, other data that may affect the daily life of patients such as body mass index, smoking and alcohol consumption should be added to this table and compared between groups, and if there is a difference, additions should be made considering the effect of these on the data.

Answer: Thank you very much for your comment. We have added other data that may affect the daily life of patients in Table S1, including body mass index, smoking history, and average daily alcohol consumption. No significant difference was found in these factors between the probiotics and placebo groups. Please see Table S1.

[3] How many of the patients had a primary diagnosis or how many had been receiving treatment for how long, The absence of any data on this is a shortcoming.

Answer: Thank you very much for your comment. I am very sorry for the lack of information about the patient's treatment history in my manuscript. In fact, our patients were recruited from the Medical Clinic of Weihai Municipal Hospital. These patients were treated with medications (such as Montelukast, Sulidie, and Symbicort Turbuhaler) for a period of 2-3 months after being hospitalized. They were recruited for this study when their medical conditions became relatively stable after receiving conventional drug treatment. Our collaborative clinical partners were responsible for randomizing patients whose symptoms were under control based on their professional

assessment into groups according to the planned setup of the current trial design. We have added this information into the manuscript. Please see line 161-164.

[4] Although the blood of the patients was taken, IgE levels were not detected?

Answer: Thank you very much for your comment. Both reviewers mentioned this important issue. Indeed, we have conducted immunological assays to detect patients' serum levels of IgG, IgM, IgA, C-reactive protein, and IgE, but most of these data were not mentioned in the first version of the manuscript due to the non-significant differences between groups. We have updated the Results and Discussion sections in the revised manuscript. Please see Figure 1a, line 338-340 and 516-523.

References

- Ou CY, Kuo HC, Wang L, Hsu TY, Chuang H, Liu CA, Chang JC, Yu HR, Yang KD. 2012. Prenatal and Postnatal Probiotics Reduces Maternal but Not Childhood Allergic Diseases: A Randomized, Double-Blind, Placebo-Controlled Trial. *Clin Exp Allergy* 42(9):1386-96.
- Joks R, Daoud A, Taningco G, Gloria CJ, Orloff K, Hammerschlag MR, Weiss S, Gelling M, Roblin PM, Nowakowski M. 2005. Minocycline treatment results in reduced oral steroid requirements and suppresses ige responses in adult asthma. *J Allergy Clin Immun* 115(2):S3.
- Cao Y, Lin SH, Zhu D, Xu F, Chen ZH, Shen HH, Li W. 2018. WeChat Public Account Use Improves Clinical Control of Cough-Variant Asthma: A Randomized Controlled Trial. *Med Sci Monit* 14(24):1524-32.
- Ashrafi MH, Xu Y, Muhamadali H, White I, Wilkinson M, Hollywood K, Baguneid M, Goodacre R, Bayat A. 2020. A microbiome and metabolomic signature of phases of cutaneous healing identified by profiling sequential acute wounds of human skin: An exploratory study. *PLoS One* 15(2): e0229545.
- Karl JP, Margolis LM, Madslie EH, Murphy NE, Castellani JW, Gundersen Y, Hoke A, Levangie MW, Kumar R, Chakraborty N, Gautam A, Hammamieh R, Martini S, Montain SJ, Pasiakos SM. 2017. Changes in intestinal microbiota composition and metabolism coincide with increased intestinal permeability in young adults under prolonged physiological stress. *Am J Physiol Gastrointest liver Physiol* 312 (6):G559-71.
- Mchardy IH, Goudarzi M, Tong M, Ruegger PM, Schwager E, Weger JR, Graeber TG, Sonnenburg JL, Horvath S, Huttenhower C, McGovern DPB, Fornace AJ, Borneman J, Braun J. 2013 Integrative analysis of the microbiome and metabolome of the human intestinal mucosal surface reveals exquisite inter-relationships. *microbiome* 1(1):17.
- Xie F, Jin W, Si H, Yuan Y, Tao Y, Liu J, Wang X, Yang C, Li Q, Yan X, Lin L, Jiang Q, Zhang L, Guo C, Greening C, Heller R, Guan LL, Pope PB, Tan Z, Zhu W, Wang M, Qiu Q, Li Z, Mao S. 2021. An integrated gene catalog and over 10,000 metagenome-assembled genomes from the gastrointestinal microbiome of ruminants. *Microbiome* 9(1):137.
- Almeida A, Mitchell AL, Boland M, Forster SC, Gloor GB, Tarkowska A, Lawley TD, Finn RD. 2019. A new genomic blueprint of the human gut microbiota. *Nature* 568(7753):499-504.
- Parks DH, Rinke C, Chuvochina M, Chaumeil PA, Woodcroft BJ, Evans PN, Hugenholtz P, Tyson

- GW. 2017. Recovery of nearly 8,000 metagenome-assembled genomes substantially expands the tree of life. *Nat Microbiol* 2(11):1533-42.
- Darzi Y, Falony G, Vieira-Silva S, Raes J. 2016. Towards biome-specific analysis of meta-omics data. *Isme J* 10:1025-8.
- Magnusdottir S, Heinken AK, Kutt L, Ravcheev D, Bauer E, Noronha A, Greenhalgh K, Jäger C, Baginska J, Wilmes P, Fleming RMT, Thiele I. 2017. Generation of genome-scale metabolic reconstructions for 773 members of the human gut microbiota. *Nat Biotechnol* 35:81-9.
- Lee-Sarwar KA, Lasky-Su J, Kelly RS, Litonjua AA, Weiss ST. 2020. Gut Microbial-Derived Metabolomics of Asthma. *Metabolites* 10(3):97.
- Depner M, Taft DH, Kirjavainen PV, Kalanetra KM, Karvonen AM, Peschel S, Schmausser-Hechfellner E, Roduit C, Frei R, Lauener R, Divaret-Chauveau A, Dalphin JC, Riedler J, Roponen M, Kabesch M, Renz H, Pekkanen J, Farquharson FM, Louis P, Mills DA, Mutius E, Ege MJ. 2020. Maturation of the gut microbiome during the first year of life contributes to the protective farm effect on childhood asthma. *Nat Med* 26(11):1766-1775.
- Platten M, Nollen EAA, Röhrig UF, Fallarino F, Opitz CA. 2019. Tryptophan metabolism as a common therapeutic target in cancer, neurodegeneration and beyond. *Nat Rev Drug Discov* 18(5): 379-401.
- Carr TF, Alkatib R, Kraft M. 2019. Microbiome in mechanisms of asthma. *Clin Chest Med* 40(1): 87-96.

August 26, 2021

Dr. Teng Ma
Inner Mongolia Agricultural University
Hohhot
China

Re: Spectrum00859-21R1 (Adjunctive probiotics alleviates asthmatic symptoms via modulating the gut microbiome and serum metabolome)

Dear Dr. Teng Ma:

Thank you for submitting your manuscript to Microbiology Spectrum. As you will see that both reviewers are quite happy with your revision. However, one did raised two minor issues. I thus would like to ask you to do a quick fix, and return the revised manuscript in less than 30 days. I will personally check if the issues are indeed fixed. If yes, we will not go another round of review since the issues are quite minor.

Please find the reviewers' comments below.

When submitting the revised version of your paper, please provide (1) point-by-point responses to the issues I raised in your cover letter, and (2) a PDF file that indicates the changes from the original submission (by highlighting or underlining the changes) as file type "Marked Up Manuscript - For Review Only". Please use this link to submit your revised manuscript. Detailed information on submitting your revised paper are below.

Link Not Available

Sincerely,

Wei-Hua Chen

Reviewer comments:

Reviewer #1 (Public repository details (Required)):

Metabolite data is necessary to submit in a public database. In addition, your provided github link is

not available.

Reviewer #1 (Comments for the Author):

Manuscript was improved after revision. Some more changes are necessary before acceptance.

1. Metabolite data is also necessary to submit in a public database. Your provided github link is not available.

2. Table S3 can be changed for clear understanding. I recommend that author summary mean value {plus minus} SD for each value and remove maximum and minimum value and present data at comparison day point in the same line (show it horizontally).

Reviewer #2 (Public repository details (Required)):

the authors give the link of deposited in a public repository in the manuscript BioProject: PRJNA722129

Reviewer #2 (Comments for the Author):

The author has appropriately explained the revisions in the manuscript. In this way, the writing is fluent and the data is in a more easily understandable format. While clinical adaptations of microbiome studies are always challenging, their data are therefore always interesting. The revisions cleared my reservations.

Preparing Revision Guidelines

- point-by-point responses to the issues I raised in your cover letter
- Upload a compare copy of the manuscript (without figures) as a "Marked-Up Manuscript" file.
- Each figure must be uploaded as a separate file, and any multipanel figures must be assembled into one file.
- Manuscript: A .DOC version of the revised manuscript
- Figures: Editable, high-resolution, individual figure files are required at revision, TIFF or EPS files are preferred

For complete guidelines on revision requirements, please see the journal Submission and Review Process requirements at <https://journals.asm.org/journal/Spectrum/submission-review-process>.

Submissions of a paper that does not conform to Microbiology Spectrum guidelines will delay acceptance of your manuscript. "

Please return the manuscript within 60 days; if you cannot complete the modification within this time period, please contact me. If you do not wish to modify the manuscript and prefer to submit it to another journal, please notify me of your decision immediately so that the manuscript may be formally withdrawn from consideration by Microbiology Spectrum.

If you would like to submit an image for consideration as the Featured Image for an issue, please contact Spectrum staff.

30 August 2021

Dear Editor:

Thank you for your and the reviewers' comments and suggestions on our manuscript. The comments and suggestions are valuable for improving our manuscript. We have read the comments carefully and revised accordingly, the revised portion of the manuscript is shown in red in the updated version of the manuscript.

We hope our revised version will now be acceptable for publication in Microbiology Spectrum, and we look forward to hearing from you. Thank you for your time and consideration.

Best regards,

Teng Ma

Answers to reviewers:

Reviewer #1:

[1] Metabolite data is also necessary to submit in a public database. Your provided GitHub link is not available..

Answer: Thank you very much for your comment. We have uploaded the serum metabolism data of asthma patients to MassIVE, which is a community resource developed by the NIH-funded Center for Computational Mass Spectrometry to promote the global, free exchange of mass spectrometry data (MSV000088033; <https://massive.ucsd.edu/ProteoSAFe/dataset.jsp?task=e185c780a113402b8b71b886223f1a22>). We have re-corrected the GitHub link, please check: <https://github.com/TengMa-Cleap/Probiotics-relieve-human-asthma-project/>. Please see line 321-325.

[2] Table S3 can be changed for clear understanding. I recommend that author summary mean value {plus minus} SD for each value and remove maximum and minimum value and present data at comparison day point in the same line (show it horizontally).

Answer: Thank you very much for your comment. We have changed Table S3 according to your suggestion, please check the supplementary material Table S3.

Reviewer #2:

[1] the authors should give the link of deposited in a public repository in the manuscript BioProject: PRJNA722129

Answer: Thank you very much for your comment, we have added a link to release metagenomic data on the NCBI website, please check <https://www.ncbi.nlm.nih.gov/bioproject/PRJNA722129/>. Please see line 320.

August 31, 2021

Prof. zhihong sun
Inner Mongolia Agricultural University
Key Laboratory of Dairy Biotechnology and Engineering, Ministry of Education, School of Food
Science and Engineering
306 Zhaowuda Road
Huhhot, Inner Mongolia 010018
China

Re: Spectrum00859-21R2 (Adjunctive probiotics alleviates asthmatic symptoms via modulating the gut microbiome and serum metabolome)

Dear Prof. zhihong sun:

Your manuscript has been accepted, and I am forwarding it to the ASM Journals Department for publication. You will be notified when your proofs are ready to be viewed.

Sincerely,

Wei-Hua Chen
Editor, Microbiology Spectrum
